# Development and Characterization of Polyester and Acrylate-Based Composites with Hydroxyapatite and Halloysite Nanotubes for Medical Applications

**DOI:** 10.3390/polym12081703

**Published:** 2020-07-29

**Authors:** Elena Torres, Ivan Dominguez-Candela, Sergio Castello-Palacios, Anna Vallés-Lluch, Vicent Fombuena

**Affiliations:** 1Textile Industry Research Association (AITEX), Plaza Emilio Sala 1, 03801 Alcoy, Spain; etorres@aitex.es; 2Technological Institute of Materials (ITM), Universitat Politècnica de València (UPV), Plaza Ferrándiz y Carbonell 1, 03801 Alcoy, Spain; ivdocan@epsa.upv.es; 3Centre for Biomaterials and Tissue Engineering, Universitat Politècnica de València (UPV), Camí de Vera s/n, 46022 Valencia, Spain; sercaspa@iteam.upv.es (S.C.-P.); avalles@ter.upv.es (A.V.-L.)

**Keywords:** biomedical polymers, hydroxyapatite, halloysite, mechanical properties

## Abstract

We aimed to study the distribution of hydroxyapatite (HA) and halloysite nanotubes (HNTs) as fillers and their influence on the hydrophobic character of conventional polymers used in the biomedical field. The hydrophobic polyester poly (ε-caprolactone) (PCL) was blended with its more hydrophilic counterpart poly (lactic acid) (PLA) and the hydrophilic acrylate poly (2-hydroxyethyl methacrylate) (PHEMA) was analogously compared to poly (ethyl methacrylate) (PEMA) and its copolymer. The addition of HA and HNTs clearly improve surface wettability in neat samples (PCL and PHEMA), but not that of the corresponding binary blends. Energy-dispersive X-ray spectroscopy mapping analyses show a homogenous distribution of HA with appropriate Ca/P ratios between 1.3 and 2, even on samples that were incubated for seven days in simulated body fluid, with the exception of PHEMA, which is excessively hydrophilic to promote the deposition of salts on its surface. HNTs promote large aggregates on more hydrophilic polymers. The degradation process of the biodegradable polyester PCL blended with PLA, and the addition of HA and HNTs, provide hydrophilic units and decrease the overall crystallinity of PCL. Consequently, after 12 weeks of incubation in phosphate buffered saline the mass loss increases up to 48% and mechanical properties decrease above 60% compared with the PCL/PLA blend.

## 1. Introduction

Tissue engineering has been exploring new methods to replace missing human tissues through biomaterials-based scaffolds, usually engineered to drive cell growth and provide shape to the creation of the new tissue. However, we should consider alternatives when selecting materials for bone fracture remodeling since conventional materials used for these applications include metallic prosthesis, bone grafts, or polymers. Currently, biopolymers are being intensively studied to replace both metal prostheses and autologous bone grafts because metal prostheses induce poor bone regeneration with formation of fragile porous bone [1] and, although autologous bone grafts induce the growth of strong bone, donor bone is needed, requiring additional chirurgical interventions, eventually causing infections [2]. Thus, polymers having easy processability to obtain desired geometries and special functionalities to accelerate bone growth will be the best option to treat bone fracture remodeling. Accordingly, biopolymers used as scaffolds for tissue engineering applications need to overcome two significant challenges. First, the biodegradation process should be controlled with non-toxic degradation by-products eliminated through natural pathways. Secondly, the material should maintain its structural and mechanical properties to avoid malformation of the new regenerated bone while healing [3].

Among all the biopolymers used for tissue engineering, bioabsorbable aliphatic polyesters are the dominant scaffolding materials because of their biodegradability properties. Biodegradable aliphatic polyesters, containing the ester functional group in their main chain, undergo hydrolytic cleavage generating oligomers, which will be subsequently assimilated into the surrounding environment. Poly (ε-caprolactone) (PCL), poly (lactic acid) (PLA) and poly (hydroxybutyrate) (PHB), approved by the U.S. Food and Drug Administration, are the most studied polyesters due to their easy processability and their tunability regarding crystallinity, thermal transition and mechanical strength properties [4,5]. The degradation rate and mechanical properties of biopolymers will be affected by the hydrophobicity, crystallinity and acidity of the selected polymer.

The hydrophobic biopolymer PCL is extensively used in drug delivery devices showing excellent biological activity. Accordingly, studies focused on its use as a scaffold and internal fixation system, although its low mechanical properties cannot meet the structural requirements of the host tissue. Consequently, an appropriate addition of fillers or blends could provide an adequate mechanical stiffness to resist in vivo stresses, preventing new tissue deformation [6,7,8]. As examples, Lowry et al. [9] tested PCL composites as internal fixation devices, observing a higher strength when using a PCL/bone complex compared with bony humerus healed with a stainless-steel implant. These observations are in concordance with the studies developed in the last decade by Rudd and co-workers [10,11,12,13,14].

Introduction of specific bioceramics can also confer new functions, such as higher biological activity. Thereupon, hydroxyapatite (HA) is broadly use as an inexpensive filler in tissue engineering [15,16,17] because of its osteoconductive properties, low inflammatory response and low toxicity in humans [18,19], based on the mineral phase of the human bone being mainly composed (around 60 wt %) of HA [20]. Therefore, introduction of HA into a polymer induces the formation of an apatite layer with similar characteristics to those of the bone mineral phase [21], inasmuch as HA improves cell attachment [22,23], inducing the differentiation of mesenchymal cells into osteoblasts, which accelerates bone formation [8]. Different authors observed both mechanical and biological improvement of biopolymer matrices with the addition of HA [24,25].

The PCL low stiffness can also be attributed to using halloysite nanotubes (HNTs) [26,27] which are an inexpensive biocompatible clay extensively used in biomedicine for drug delivery due to their tubular shape. HNTs also support cell adhesion, ascribed to HNTs surface nano-roughness, which acts as an anchor frame [28], and the interaction between silanol groups present on the HNTs surface [29] with hydroxyl and amino groups present on proteins.

In a previous study [30], mechanical and thermal properties of PCL were studied by modifying the additive percentage of the bioactive fillers HA and HNTs. Accordingly, the additive threshold was stablished in 7.5 wt % of HNTs and 20 wt % of HA achieving a noticeable improvement in mechanical properties with the simultaneous addition of the two fillers. As a result, the flexural modulus improved up to 112.3% reaching values of 886.8 MPa (standard deviation = 42.1), and Young’s modulus increased to 109.3% with its greatest value at 449.6 MPa (standard deviation = 17.12). Knowing that HA promotes the formation of a layer of new bone, and that HA and HNTs alter hydrophobicity behavior, in a second study, [23], biological properties such as cell viability, proliferation and morphology supplied by both fillers were studied and compared on different pairs of polymers with similar chemical nature but different hydrophobicity. Accordingly, the hydrophobic polyester PCL was modified when it blended with poly (lactic acid (PLA) and combined with HA nanoparticles and HNTs. However, the hydrophilic poly (2-hydroxyethyl methacrylate) (PHEMA) was copolymerized as monomer with ethyl methacrylate (EMA) and also combined with HA and HNTs. These polymers, although dissimilar to PCL and PLA in terms of chemical nature and biodegradability, were chosen for comparison purposes because they are used for hard tissue applications. Initially, the in vitro biological development of polymers with different hydrophobicity showed that cells preferably proliferate on moderately hydrophobic surfaces (PCL/PLA). However, over longer culture periods, cell proliferation increased on more hydrophilic materials (P(HEMA-co-EMA)). Inorganic nanoparticles (HA and HNTs) improve cell viability and proliferation compared to the raw materials. We assumed that reduced cell spreading on hydrophobic surfaces at long culture times might occur as a consequence of two effects: protein absorption competition and the steric hindrance effect (solvation).

We acknowledge that the contributions of the previous studies [23,30] need to be accomplished by monitoring the degradation rate of biodegradable polyesters modified with HA and HNT. Biomedical polymers, after implantation, undergo significant changes regarding mechanical properties influenced by their degradation process. Considering that 75% of the human body is composed of water, hydrolytic degradation of aliphatic polyesters is an interesting feature for tissue engineering materials. Bone remodeling implies time-limited applications, which requires the elimination or degradation of the biopolymer after use to restore the surrounding living medium.

For all the above-mentioned reasons, we studied the bioactivity of polyester (PCL, PCL/PLA and PLA) and acrylates (PHEMA, P(HEMA-co-EMA) sets and their HA- and HNT-based nanocomposites, as well as the degradability of the polyester-based set. To determine the correct distribution of the fillers, a study was conducted using SEM-EDS and an evaluation of their wettability by measuring the contact angle. Finally, to demonstrate if the loads introduced in the nanocomposites diffuse to the environment, we evaluated the mechanical properties of the nanocomposites using tensile and flexion tests.

## 2. Experimental

### 2.1. Materials

Poly (ε-caprolactone (PCL), with trade name CAPA 6500, was provided by Solvay Interox (Solvay Interox, Warrington, UK). CAPA 6500 is a high-molecular-weight thermoplastic linear polyester derived from its own lactone monomer. PLA Ingeo™ biopolymer 6201D is a thermoplastic available in pellet form with a glass transition temperature of 55–60 °C and a melting point of 155–170 °C. NatureWorks LLC (Nature Works LLC, Minnetonka, MN, USA). Hydroxyapatite (HA) with chemical formula (HCa_5_O_13_P_3_), halloysite nanotubes (HNTs) (Al_2_Si_2_O_5_ (OH)_4_ 2H_2_O), ethyl methacrylate (EMA) with 99% purity, and hydroxyl-2-ethyl methacrylate (HEMA), with a minimum of purity of 96%, were supplied by Sigma-Aldrich (Madrid, Spain). Benzoin and ethylene glycol dimethacrylate 98% (EGDMA) were used as ultraviolet initiator and crosslinking agent during the preparation of HEMA/EMA compounds. Both were also supplied by Sigma Aldrich.

### 2.2. Preparation of the Polymer-Based Hybrids

The first set of materials based on PCL and PLA were received in pellet form and dried prior to their preparation in an air oven at 50 and 60 °C, respectively, to remove humidity. In parallel, HA and HNTs were dried separately in a vacuum oven for 48 h at 200 and 80 °C. The proportions detailed in Table 1 were weighed and pre-mixed in a zipper bag. By a twin screw co-rotating extruder with different temperature profiles, the mixtures were mechanically homogenized. Specifically, for PCL-based compounds, the temperature profile was 65/75/85/90 °C, and for PLA based-compounds, we used temperatures of 170/173/17/180 °C. After the extrusion process, the samples were cooled to room temperature and pelletized. Again, prior to the injection process, the different compounds were dried under the same conditions as mentioned above. The injection was carried out in a Meteor 270/75 injection molding machine (Mateu and Solé, Barcelona, Spain) using as temperature profiles of the extruder: 80/80/85/85/90 °C for PCL compounds and 170/173/175/180 °C for PLA compounds. Next, 13 mm diameter samples were punched out.

The second set of compounds, based on ethyl methacrylate (EMA) and hydroxyl-2-ethyl methacrylate (HEMA), was obtained by simultaneous polymerization of the monomers, as summarized in Table 1. Using a 1:1 monomer ratio between HEMA and EMA to obtain the copolymer, the mixtures were stirred with 1 wt % benzoin and 0.5 wt % EGDMA. The corresponding ratios of HA and HNTs were added and stirred for 15 min. Each mixture was injected into a glass template for polymerization in an ultraviolet oven for 24 h and subsequently a 24 h post-polymerization process in an oven at 90 °C was required. The samples were immersed in boiling ethanol and cut into 13 mm diameter samples.

### 2.3. Contact Angle Measurements

The water contact angles (WCAs) of the nanocomposites were measured on the surface of the dry samples in the sessile drop mode. An Easy Drop Standard goniometer model FM140 (110/220 V, 50/60 Hz) supplied by Krüss GmbH (Hamburg, Germany) was used for this purpose. To determine the water contact angle, we used the Drop Shape Analysis SW21 (DSA1) software. A minimum of five replicates of each sample were analyzed, yielding a standard deviation of less than 5%.

### 2.4. Mechanical Properties

Tensile properties of PCL/PLA blends loaded with HA and HNTs were obtained using a universal test machine (Ibertest ELIB 30, SAE Ibertest, Madrid, Spain) according to ISO 527. Assays were carried out with a 5 kN load cell and a crosshead speed of 10 mm·min^−1^. Moreover, to determine the Young’s modulus more accurately, an axial extensometer IB/MFQ-R2 from Ibertest (Madrid, Spain) coupled to the universal test machine was used. The Young’s modulus was calculated in each case from the stress–strain initial slope and averaged from five replicates.

### 2.5. Hydroxyapatite Nucleation

Hydroxyapatite nucleation was followed on three replicates per sample and time point. First, a simulated body fluid (SBF) solution with an ion concentration close to that of human blood plasma was prepared by the method proposed by Kokubo and coworkers [31,32]. To obtain the SBF, we prepared two solutions. Solution 1 consisted of 1.599 g of NaCl (Scharlau, 99% pure), 0.045 g of KCl (Scharlau 99% pure, Barcelona, Spain), 0.110 g of CaCl_2_·6H_2_O (Fluka 99% pure, Madrid, Spain) and 0.061 g of MgCl_2_·6H_2_O (Fluka) in deionized ultrapure water (Scharlau) up to 100 mL. Solution 2 was prepared by dissolving 0.032 g of Na_2_SO_4_·10H_2_O (Fluka), 0.071 g of NaHCO_3_ (Fluka) and 0.046 g of K_2_HPO_4_·3H_2_O (Aldrich, 99% pure) in water up to 100 mL. Both solutions were buffered at pH 7.4 by adding the necessary amounts of aqueous 1 Mtris-hydroxymethyl aminomethane, (CH_2_OH)_3_CNH_2_ (Aldrich), and 1 M hydrochloric acid (HCl, Aldrich, 37% pure). Next, both solutions were mixed to obtain SBF with the following molar ion concentrations: 142 Na^+^, 5.0 K^+^, 1.5 Mg^2+^, 2.5 Ca^2+^, 148.8 Cl^−^, 4.2 HCO_3_^−^, 1.0 HPO_4_^2−^ and 0.5 SO_4_^2−^ mM. Samples were immersed in individual vials containing 10 mL of SBF solution with hydrazine (NaH_2_) to prevent bacterial proliferation. The vials were placed in an incubator at 37 °C and 5% CO_2_. A set of samples were withdrawn after 7 and 14 days.

### 2.6. Cell Seeding

NIH 3T3 fibroblast cells were expanded in the presence of 4.5 g L^−1^ glucose supplemented with 10% fetal bovine serum (Thermo Fisher, Gibco, Waltham, MS, USA) and 1% penicillin/streptomycin (P/S; Thermo Fisher, Gibco) in Dulbecco’s modified Eagle medium (DMEM; Thermo Fisher, Gibco) at 37 °C in a 5% CO_2_ incubator until confluence. After reaching confluence (3 days), cells were withdrawn from the culture flask. To proceed, 5 mL of versene solution (0.48 mM) formulated in 0.2 g ethyldiaminotetraacetic acid (EDTA) per liter of phosphate buffered saline (PBS) supplied by ThermoFisher (Gibco), were added for 5 min at 37 °C, and then removed. After, to neutralize the versene solution, 10 mL of DMEM was added, and the suspensions were centrifugated at 1000 rpm for 5 min. Then, the cells were resuspended in 1 mL medium, counted, diluted and seeded on the samples at a density of 2 × 10^4^ cells cm^−2^.

### 2.7. Morphological Analysis

A ZEISS FESEM ULTRATM 55 scanning electron microscopy (SEM) device was used to analyze the morphology of the HA coatings and the NIH 3T3 fibroblast cells and their layout on the surfaces. The morphology of the HA coatings was studied by SEM and energy-dispersive X-ray spectroscopy (EDS) images obtained to validate the formation of a hydroxyapatite layer and the Ca/P ratio. To this end, the samples were sputter-coated with carbon under vacuum through a BALL-TEC/SCD 005 sputter coater. The mapping spectra were taken at 15 kV of acceleration voltage and 5 mm working distance; a secondary electron detector was used. Silicon was used as optimization standard. The mappings were taken at a magnification of 5000×.

In the study of NIH 3T3 fibroblast cells, arrangements were analyzed after 1 and 14 days of incubation. After each period, the culture medium was removed to rinse the samples in phosphate buffer (PB; Affymetrix, Santa Clara, CF, USA) and samples fixed with 4% paraformaldehyde solution during 30 h at 37 °C. A vacuum system was used to remove the water and to avoid any deformations on cell morphology. For this purpose, samples were rinsed in PBS twice and carefully frozen in liquid nitrogen and transferred to a freezer-dryer for drying.

### 2.8. Degradation of PCL and PCL/PLA Based Hybrids

Degradation of PCL and PCL/PLA loaded with HA and HNTs was followed in vitro at 37 °C using PBS (0.01 M (NaCl 0.138 M; KCl 0.0027 M) with a pH 7.4, at 25 °C was supplied by Sigma Aldrich). Due to the stability of thermostable compounds based on PHEMA, this study was only carried out on compounds based on PCL. With the aim of accelerating the process, samples were previously immersed in a 2M NaOH solution for 24 h. Three replicates of each composition were immersed in individual tubes with 10 mL of Dulbecco’s Phosphate Buffered Saline (DPBS, Sigma Aldrich) (pH 7.4) with screw caps and maintained at 37 °C in an incubator. Each sample was removed after 4, 8 and 12 weeks, rinsed thoroughly with deionized water, and dried in an oven at 35 °C for 12 h.

The weight loss and the mechanical integrity of the materials were evaluated. An electronic balance with a resolution of 0.1 mg was used to determine the mass loss as follows:(1)$$\%\;Mass\;loss=\frac{Mi-Mf}{Mi}\times 100$$
where *Mi* is the initial mass and *Mf* is the final mass of the dry sample.

## 3. Results and Discussion

The water contact angle formed in the range of 40°–70° on a polymeric surface is known to influence cell attachment, since chemical surface interactions are a key factor during the bio-adhesion process [33]. Polymers with contact angles in this range, with a different chemical nature, were thus selected for this analysis, and their hydrophilicity was slightly modified by blending or copolymerizing with others of the same family. Therefore, polymer chemical surfaces were modified by blending hydrophilic/hydrophobic polymers and/or filling the polymer matrix with HA or HNT to analyze their role in wettability.

From the results in Figure 1, we can discern that the most hydrophobic sample was PCL with a contact angle of 105°. Blending PCL with the more hydrophilic PLA, the surface wettability improved 25.3%, with a value of 83.7°. Conversely, PHEMA was at the hydrophilic end, with a contact angle of 59.7°, and the copolymerization with the more hydrophobic EMA decreased surface wettability up to 73.7°. The addition of HA and HNTs clearly improved surface wettability on neat samples (PCL and PHEMA), but this effect was scarcely observed in mixed samples (PCL/PLA and P(HEMA-co-EMA)), considering the standard deviation. This effect could be attributed to the intermediate wettability of these samples, with contact angles in the vicinities of 80°, together with their heterogeneous composition at the nanoscale.

After determining the wettability of the samples and the influence of the addition of HA and HNTs fillers, a SEM-EDS analysis was conducted to firstly determine if the loads were homogeneously dispersed and, secondly, to assess the formation of a hydroxyapatite layer resulting from the incubation in SBF at 37 °C. In addition, the EDS analysis (EDS spectra not shown) allowed the quantification of the Ca/P ratio and the comparison with that of the stoichiometric HA (Ca_10_(PO_4_)_6_(OH)_2_), Ca/P = 1.67 [34].

Figure 2 shows the images obtained after 7 and 14 days in SBF on the samples based on PCL/PLA. After seven days, the PCL-based hybrids did not efficiently induce apatite growth. Precipitation on PCL and PCL/PLA compounds without needle conformation corresponded to the salt dissolved in SBF medium, usually NaCl. The samples with HA filler provided nucleation sites, and the silanol groups (Si–OH) present in HNTs provide favorable locations for apatite nucleation. We speculated that the electrostatic interaction drives the formation of calcium silicate [35], since comparing pure polymers (especially PCL and PCL/PLA blends) with HA- and HNTs-modified materials, the nucleation efficiency increases with the filler. We already showed that both polar carboxyl groups and hydroxyl groups induce apatite nucleation [36].

However, Figure 3 summarizes SEM images of samples based on PHEMA/EMA. The polymethacrylate-based hybrids induced efficient apatite growth, with the exception of pure PHEMA, on which only scattered precipitated salts were observed on the surface. On the contrary, the P(HEMA-co-EMA) surface showed plenty of precipitates forming large and clear cauliflowers with intricate needle-shaped crystals [36]. As observed, once apatite nucleates on a location, it grows radially outward [37], creating cauliflower or hemispherical structures combined to form a continuous layer. Due to the weak hydrophilicity of P(HEMA-co-EMA), the biological activity is higher than PHEMA, where the number of polar groups available for nucleation per unit volume on the surface is greater. Therefore, the P(HEMA-co-EMA) surface adsorbs Ca^2+^ ions from the SBF solution more efficiently, thereby increasing the concentration of Ca^2+^ ions on the surface, and also forming Ca–P nucleation sites [38]. The first layer of apatite molecules generates the cauliflower aggregates from the secondary nucleation, observed especially in P(HEMA-co-EMA). This process induces a spherical growth perpendicular to the surface structure, which leads to the formation of clusters or grape-like structures [21].

Figure 4 reveals the assessment of the Ca/P ratio to verify the formation of hydroxyapatite layers. In most determinations, the Ca/P atomic ratio remained in acceptable values between 1.3 and 2, which pointed to the formation of calcium and phosphate deposits resembling physiological apatite structures [34]. Particularly, the highest Ca/P ratios were obtained in the PCL, PHEMA and PHEMA 20HA samples after an incubation period of seven days, all with values higher than 1.8. After 14 days of incubation, the Ca/P ratio tended to the physiological ratio of 1.67 in most samples. However, PHEMA did not show calcium on the surface after 14 days of incubation. The hydrophilic character of this polymer probably hinders the deposition of salts on its surface and their evolution toward HA. This observation coincides with that of [36], where its closely-related poly (hydroxyethyl acrylate) (PHEA) did not induce an efficient apatite growth, whereas its copolymer with EMA did.

In the SEM morphological images (Figure 5), we see the in vitro biological development of polymers with different hydrophobicity. At initial stages (day 1), cells preferably proliferated and colonized moderately on the hydrophobic surface (PCL/PLA, with contact angle of 83.7°, as summarized in Figure 1). Thus, cells appear round, where interactions occur primarily between them or with the extracellular matrix, thus resulting in a monolayer of cells with few bonding sites with the polymer surface. When longer incubation periods were analyzed (14 days), cell proliferation was favored in more hydrophilic polymers, as is the case of P(HEMA-co-EMA), with a contact angle of 73.7°. In these samples, cells exhibited a flatter morphology, establishing contact with the polymer surface. With the addition of HA and HNTs inorganic fillers, in general terms, we observed an increase in proliferation compared with the raw materials. As different authors have concluded, this improvement can be attributed to the generation of new reactive sites with Ca^2+^ and PO_4_^3−^ groups present in HA that bind with negative carboxylate and positive amino groups in proteins, respectively [39,40,41,42]. However, due to the presence of silanol groups (Si–OH) located at the surface of HNTs, the formation of hydrogen bonds between HNTs and proteins is allowed [29]. The results showed a greater proliferation at initial stages on moderately hydrophobic polymers (PCL/PLA), whereas over longer culture periods, more hydrophilic polymers P(HEMA-co-EMA) seem to improve cell proliferation, in concordance with the results by Zhou et al. [42]. Using polyvinyl alcohol (PVA), a highly hydrophilic polymer, the authors concluded that highly polar OH groups on neat PVA films might account for the delayed attachment of bone cells. Thus, we can possibly assume that reduced cell spreading on hydrophobic surfaces at long culture times might occur as a consequence of the protein absorption competition and the steric hindrance effect (solvation).

Regarding the mechanical properties, PLA is a hard polymer with good mechanical properties, its brittleness being its main disadvantage. However, PCL is a very soft polymer with a slow degradation rate. After mixing PLA and PCL, the synergy results in retaining the advantages of each polymer. Compared to pure PLA, the mixture has greater flexibility, hydrophobicity and crystallinity, which translate into a slower degradation rate (several months to years) [43]. To study if the PCL crosslinking factor (crystallinity) limits the mobility of the chains and, therefore, the degradation rate, an analysis of the mass loss and mechanical properties after 4, 8 and 12-weeks of incubation in PBS was conducted. The PCL crystallinity tends to reduce the chain mobility and thus its hydrolysis by means of the hindered access of enzymes to the polymer matrix [23].

From the results shown in Figure 6, the PCL-based material did not show significant mass loss after degradation in PBS at 37 °C for 12 weeks. Although the mass loss of all PCL-based samples gradually increased, compared with samples containing inorganic fillers, the mass loss of pure PCL increased rapidly during the 12-week degradation process. The lower mass loss was observed in samples with HA because HA contains hydroxyl groups, which can neutralize the medium by reacting with degraded acid by-products, thereby reducing the effect of acid catalysis on PCL hydrolysis [44]. Conversely, PCL/PLA blends showed a maximum mass loss rate of the samples when using the two fillers, 41% after 12 weeks. Blending PCL with PLA provides hydrophilic units and reduces the overall crystallinity of PCL, thereby improving the accessibility of water molecules and ester bonds, and increasing the hydrolysis rate [45]. The addition of fillers provides hydrophobicity, and the nano-roughness of the surface promotes interaction with water molecules and thus causes hydrolytic cracking. Therefore, the evolution of mass loss is related to the mechanical properties, which were followed through the analysis of their Young’s moduli, gathered in Figure 7.

The Young’s modulus of the PCL samples was about 275 MPa. We observed that as the incubation time and the percentage of mass loss increased, the mechanical properties gradually decreased. The neat PCL samples showed a sharp decrease in Young’s modulus, but those with HA retained their Young’s modulus due to the filler reinforcement. However, the samples with HA and HNT showed a gradual decrease in mechanical properties over time, since the additional threshold of the nanoloads could be exceeded as the polymer degraded, and the agglomerates could act as weak points and failure initiation points. Blending PCL with PLA resulted in the Young’s modulus increasing to 800 MPa. In PCL/PLA blends with the presence of inorganic fillers, the faster degradation rate, associated with more hydrophilic properties and more amorphous phases, led to a faster decline in mechanical properties. More precisely, the PCL/PLA 20HA 7.5 HNTs blend provided a Young’s modulus 60% lower than that of the PCL/PLA blend.

## 4. Conclusions

Studying the influence of the HA and HNTs fillers on the wettability of the selected polyesters and acrylates, we measured water contact angles, which corroborated that PCL is the most hydrophobic sample, whereas PHEMA is hydrophilic. The addition of HA and HNTs clearly improved the surface wettability of neat samples, but not of the binary samples (PCL/PLA and P(HEMA-co-EMA)).

Secondly, distribution of the fillers into the polymer matrices was studied through EDS mapping. A homogeneous distribution of HA was observed on all polymers, with the exception of PHEMA. Nonetheless, the presence of HNTs yields large aggregates on more hydrophilic polymers, derived from the hydrophobic character of the nanotubes, resulting in a good dispersion in non-polar polymers and creating agglomerations in hydrophilic polymers due to the lower interfacial adhesion. Considering HA nucleation on polymer surfaces, hydrophobic polymers did not show an efficient induction of apatite growth. Conversely, large hydroxyapatite cauliflower-shaped crystals formed on moderately hydrophilic surfaces. We discovered that PHEMA is excessively hydrophilic, so promotes HA nucleation on its surface.

Finally, the evaluation of the degradation rate of the biodegradable PCL-based samples demonstrated that both the blending PCL with PLA and the addition of HA and HNTs provide hydrophilic units, as well as decrease the overall crystallinity of PCL, thereby improving the accessibility of water molecules to ester linkages and thus the hydrolytic cleavage. Consequently, the faster degradation rate and reduced mass lead to a faster drop of mechanical properties.

## Figures and Tables

**Figure 1 polymers-12-01703-f001:**
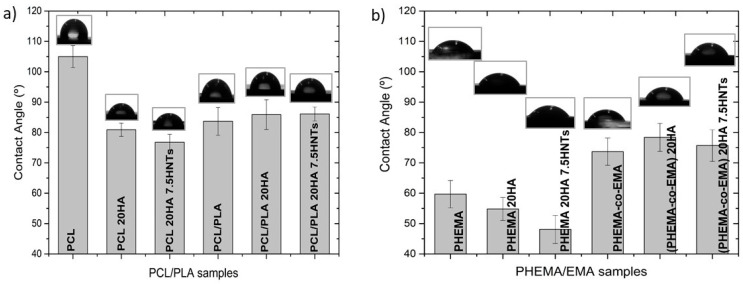
Variation of contact angle in compounds based on (**a**) PCL/PLA and (**b**) PHEMA/EMA.

**Figure 2 polymers-12-01703-f002:**
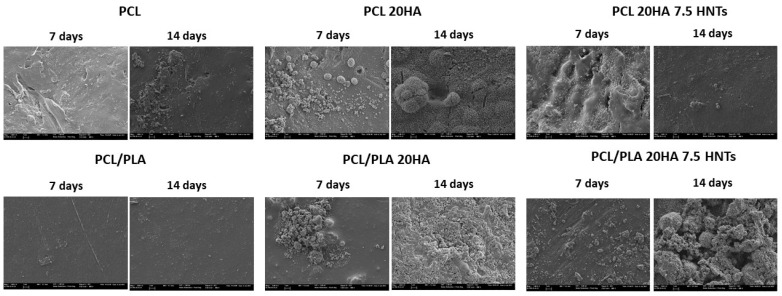
SEM images (5000×) taken for hydroxyapatite nucleation analysis on samples based on PCL/PLA.

**Figure 3 polymers-12-01703-f003:**
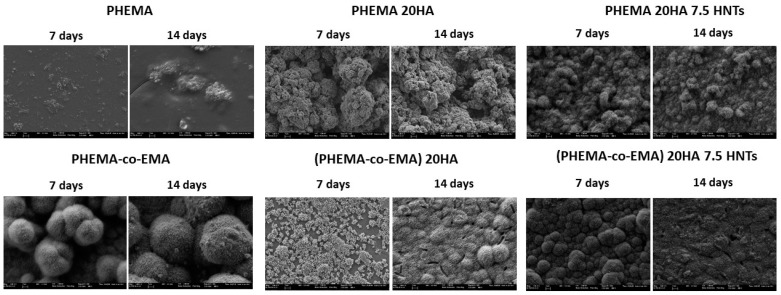
SEM images (5000×) taken for hydroxyapatite nucleation analysis on samples based on PHEMA/EMA.

**Figure 4 polymers-12-01703-f004:**
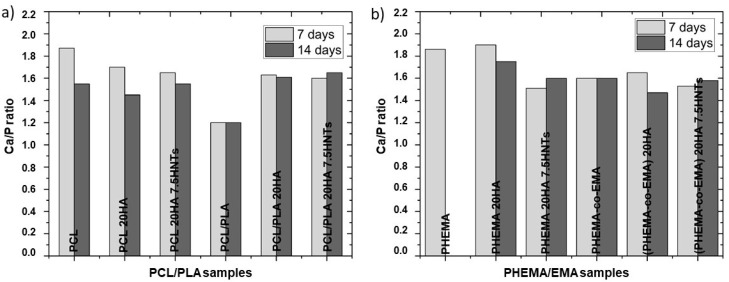
Ca/P ratio after 7 and 14 days of incubation in SBF obtained by EDS on nanocomposites based on: (**a**) PCL and PCL/PLA, (**b**) PHEMA and P(HEMA-co-EMA).

**Figure 5 polymers-12-01703-f005:**
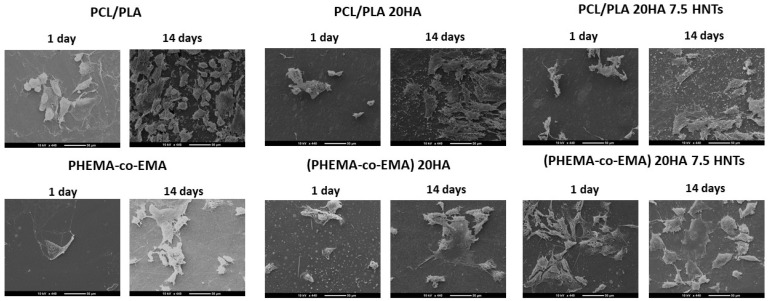
SEM images (400×) taken for cell proliferation at 1 and 14 days on samples based on PCL/PLA and PHEMA/EMA.

**Figure 6 polymers-12-01703-f006:**
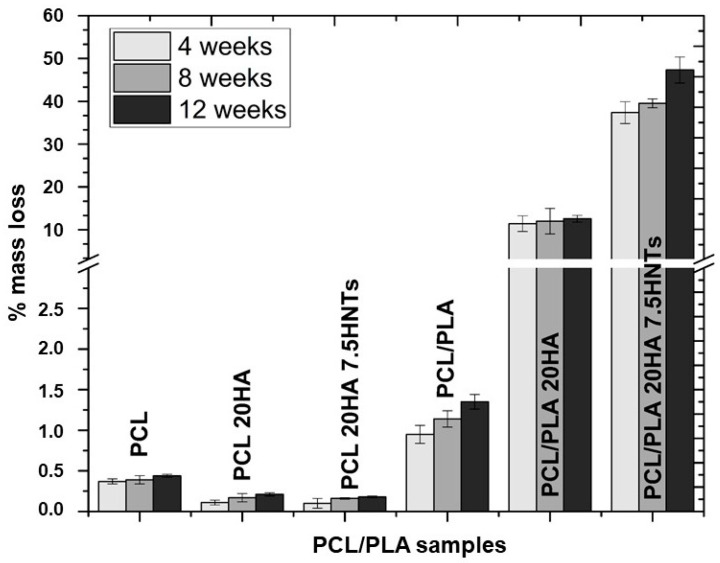
Results of degradation: mass loss (%) after 4, 8 and 12 weeks for PCL-based materials.

**Figure 7 polymers-12-01703-f007:**
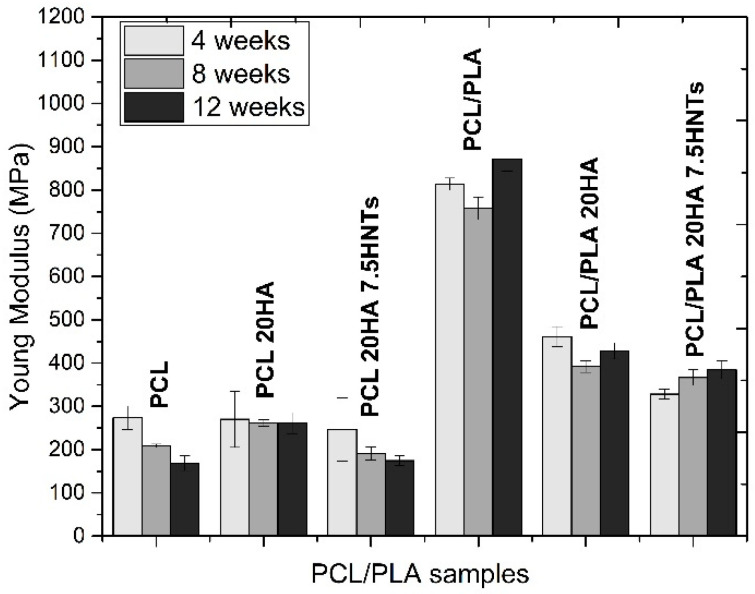
Results of Young modulus following 4, 8 and 12 weeks of degradation for PCL/PLA-based materials.

**Table 1 polymers-12-01703-t001:** Composition and coding of PCL, PCL/PLA, PHEMA, and P(HEMA-co-EMA) composites.

Code	Composition (wt %)			
	PCL	PLA	HA	HNTs
PCL	100	-	-	-
PCL_20HA	80	-	20	-
PCL_20HA_7.5HNTs	72.5	-	20	7.5
PCL/PLA	50	50	-	-
PCL/PLA_20HA	40	40	20	-
PCL/PLA_20HA_7.5HNTs	36.25	36.25	20	7.5
Code	Composition (wt %)			
	HEMA	EMA	HA	HNTs
PHEMA	100	-	-	-
PHEMA_20HA	80	-	20	-
PHEMA_20HA_7.5HNTs	72.5	-	20	7.5
P(HEMA-co-EMA)	50	50	-	-
P(HEMA-co-EMA)_20HA	40	40	20	-
P(HEMA-co-EMA)_20HA_7.5HNTs	36.25	36.25	20	7.5

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
