# Peer review of "Development and Characterization of Polyester and Acrylate-Based Composites with Hydroxyapatite and Halloysite Nanotubes for Medical Applications"

_polymers, 2020, doi:10.3390/polym12081703_

Round 1

Reviewer 1 Report

Two classes of multicomponent biocomposites on the base of the hydrophobic polyesters (PCL, PLA) filled by hydroxyapatite and halloysite nanotubes and hydrophilic polyester (PHEMA) with the same fillers were explored as the biomedical material to regulate mechanical and degradation behavior. So, the authors propose generally speaking two different in hydrophobicity and sophisticated biomaterials. Embedding the modifiers makes the surface of biopolyesters more hydrophilic (the contact angle data) and accelerate/restrict the degradability of PCL-PLA blends.

To achieve the main conclusions E. Torres and co-authors used a variety of experimental techniques such as SEM morphological analysis, EDS atomic content evaluation, contact angle registration, mechanical measurements, proliferated cell seeding, and others. The structure of the manuscript is logically reasonable. The introductory section provides an appropriate background of the manuscript topics that immediately acquaints the reader with the problem of biopolymer modification. The literature cited is quite relevant to this study. The characteristics of the composites have presented in the submission are fairly coherent and cognitive for the experts in the area of biomedicine material science.

Several remarks should be pointed out

A few words about the idea of publication: why the authors have chosen for comparison two opposite in the hydrophobic-hydrophilic balance composites (PLA/PCL and polyacrylates) but simultaneously dramatically dissimilar in (bio)degradation.

It is worth to recommend to insert the abbreviations (PCL, PLA, PHMA..) immediately in the Abstract.

L82,83: If the p-values are the characteristics of statistical significance within 0 < p < 1., what mean the equalities in the parentless?  

Please reformulate the phrase in Line 350. What does the fragment “the hydrophobic end” mean?  At least, you should mention that you say about the scale or spectrum of hydrophilicity, namely, the end of what?

Sometimes, the authors express the sentences in a somewhat wordy manner e.g  “The completion…could be accomplished” (L 99), please edit to go to a laconic manner.

All flaws belong to a minor score area and after correction, the manuscript can be promoted for the publication

Reviewer 2 Report

The manuscript under evaluation would be suitable for publication after revision. Authors should evaluate the molar mass changes at least presenting the GPC traces during the degradation process.
